# Moment Matching Denoising Gibbs Sampling

**Mingtian Zhang**[*]
Centre for Artificial Intelligence
University College London
m.zhang@cs.ucl.ac.uk

**Alex Hawkins-Hooker**
Centre for Artificial Intelligence
University College London
a.hawkins-hooker@cs.ucl.ac.uk

**Brooks Paige**
Centre for Artificial Intelligence
University College London
b.paige@ucl.ac.uk

**David Barber**
Centre for Artificial Intelligence
University College London
david.barber@ucl.ac.uk

## Abstract

Energy-Based Models (EBMs) offer a versatile framework for modeling complex data distributions. However, training and sampling from EBMs continue to pose significant challenges. The widely-used Denoising Score Matching (DSM) method [41] for scalable EBM training suffers from inconsistency issues, causing the energy model to learn a 'noisy' data distribution. In this work, we propose an efficient sampling framework, (pseudo)-Gibbs sampling with moment matching, which enables effective sampling from the underlying clean model when given a 'noisy' model that has been well-trained via DSM. We explore the benefits of our approach compared to related methods and demonstrate how to scale the method to high-dimensional datasets.

## 1 Energy-Based Models

Energy-Based Models (EBMs) have attracted a lot of attention in the generative model literature [25, 46, 7, 35]. EBMs are a type of non-normalized probabilistic model that determines the probability density function without a known normalizing constant. For continuous data $x$, the density function of an EBM is specified as $q_\theta(x) = \exp(-f_\theta(x))/Z(\theta)$ where the $f_\theta(x)$ is a nonlinear function with parameter $\theta$ and $Z(\theta) = \int \exp(-f_\theta(x))\,\mathrm{d}x$ is the normalization constant that is independent of $x$. The energy parameterization allows for greater flexibility in model parameterization and the ability to model a wider range of probability distributions. However, the lack of a known normalizing constant makes training these models challenging. We start by giving a brief introduction of how to estimate $\theta$ in EBMs and refer the reader to [37] for a detailed overview of different training techniques for continuous EBMs.

**Likelihood-based training:** A classic method to learn $\theta$ is to minimize the KL divergence between the data distribution $p_d$ and the model density $q_\theta$, which is defined as

$$\mathrm{KL}(p_d||q_\theta) \doteq - \int p_d(x) \log q_\theta(x)\,\mathrm{d}x \doteq - \int p_d(x) f_\theta(x)\,\mathrm{d}x - \log Z(\theta), \tag{1}$$

where we use $\doteq$ to denote the equivalence up to a constant that is independent of $\theta$. The integration of $p_d(x)$ can be approximated by Monte Carlo with the training dataset $\mathcal{X}_{train} = \{x_1, \cdots, x_N\} \sim p_d(x)$; in this case, it is equivalent to the maximum likelihood estimate (MLE) [5]. However, for EBMs, minimizing the KL divergence requires the estimation of $Z(\theta)$, which is intractable for nonlinear $f_\theta(x)$ defined by a neural network. Various methods have been proposed to alleviate the

---

[*]This work was partially done during an internship in Huawei Noah's Ark Lab.

37th Conference on Neural Information Processing Systems (NeurIPS 2023).

intractability by introducing techniques like Markov chain Monte Carlo (MCMC) [13, 26, 8, 11] or adversarial training [18, 47, 6].

**Score-based training:** Alternatively, [16] proposes to minimize the Fisher divergence to learn $\theta$, which is defined as

$$\mathrm{FD}(p_d||q_\theta) = \frac{1}{2} \int p_d(x) ||s_{p_d}(x) - s_{q_\theta}(x)||_2^2 \, \mathrm{d}x, \tag{2}$$

where we use $s_p(x)$ to denote the score function of distribution $p$: $s_p(x) \equiv \nabla_x \log p(x)$. Under certain regularity conditions, the Fisher divergence is equivalent to the score-matching (SM) objective [16],

$$\mathrm{FD}(p_d||q_\theta) \doteq \frac{1}{2} \int p_d(x) \left( ||s_{q_\theta}(x)||_2^2 + 2 \mathrm{Tr}(\nabla_x s_{q_\theta}(x)) \right) \mathrm{d}x, \tag{3}$$

which does not require estimation of the intractable $Z(\theta)$. However, this objective needs to calculate the Hessian trace $\nabla_x s_{q_\theta}(x) = -\nabla_x^2 f_\theta(x)$ in every gradient step during training, which is computationally expensive and does not scale to high dimensional data or requires approximation [38]. In this paper, we will focus on another training method, denoising score matching [41], which overcomes the tractability and scalability issues mentioned above, and is introduced in the next section.

## 1.1 Denoising Score Matching

For the target data density $p_d(x)$, a noise distribution $p(\tilde{x}|x) = \mathcal{N}(x, \sigma^2 I)$ is introduced to construct a noised data distribution $\tilde{p}_d(\tilde{x}) = \int p_d(x)p(\tilde{x}|x) \, \mathrm{d}x$. Denoising score matching (DSM) [41] minimizes the Fisher divergence between the noised data distribution $\tilde{p}_d(\tilde{x})$ and an energy-based model $\tilde{q}_\theta(\tilde{x}) = \exp(-f_\theta(\tilde{x}))/Z(\theta)$, with

$$\mathrm{FD}(\tilde{p}_d||\tilde{q}_\theta) = \frac{1}{2} \int \tilde{p}(\tilde{x}) ||s_{\tilde{p}_d}(\tilde{x}) - s_{\tilde{q}_\theta}(\tilde{x})||_2^2 \, \mathrm{d}\tilde{x}$$

$$\doteq \frac{1}{2} \iint p(\tilde{x}|x)p_d(x) ||\nabla_{\tilde{x}} \log p(\tilde{x}|x) - s_{\tilde{q}_\theta}(\tilde{x})||_2^2 \, \mathrm{d}\tilde{x} \, \mathrm{d}x$$

$$\doteq \frac{1}{2} \iint p(\tilde{x}|x)p_d(x) \left\| \frac{\tilde{x} - x}{\sigma^2} + s_{\tilde{q}_\theta}(\tilde{x}) \right\|_2^2 \, \mathrm{d}\tilde{x} \, \mathrm{d}x \,, \tag{4}$$

where the last equation is due to $\nabla_{\tilde{x}} \log p(\tilde{x}|x)$ being tractable for the Gaussian distribution $p(\tilde{x}|x)$.

Compared to the KL or SM objectives, the DSM objective is scalable and well-defined when the data distribution is singular[2] [49] and can alleviate the blindness problem of score matching [35, 45, 51]. On the other hand, there is a notable disadvantage associated with the DSM objective: for a fixed $\sigma > 0$, the DSM objective is not a consistent objective for learning the underlying data distribution $p_d$ since $\mathrm{FD}(\tilde{p}_d||\tilde{q}_\theta) = 0 \implies \tilde{p}_d = \tilde{q}_\theta \neq p_d$. A common solution is to anneal $\sigma \to 0$ during training. However, Equation 4 is not defined when $\sigma = 0$ since the division in Equation 4 will make $(\tilde{x} - x)/\sigma^2$ unbounded, which results in an inconsistent objective. Annealing $\sigma$ increases the variance of the training gradients [37, 42], which makes the optimization challenging in practice.

To overcome the challenges, we propose an alternative data generation scheme: we use DSM with a fixed $\sigma > 0$ to train a 'noisy' energy model and then construct a sampler which targets the underlying 'clean' model. Specifically, our contributions are summarized as follows:

- We demonstrate that for an EBM that learns a noisy data distribution, there exists a unique underlying clean model which recovers the true data distribution.
- We introduce a pseudo-Gibbs sampling scheme incorporating an analytical moment-matching approximation of the denoising distribution. This allows us to sample from the underlying clean model without requiring additional training.
- We illustrate how to scale our method for high-dimensional data and demonstrate the generation of high-quality images using only a single level of fixed noise. Furthermore, we showcase the application of our proposed method in multi-level noise scenarios, closely resembling a diffusion model.

---

[2]The singular distribution is not absolutely continuous (*a.c.*) with respect to the Lebesgue measure, thus doesn't allow a density function [40, p.172]. A typical example is a data distribution supported on a lower-dimensional manifold. In this case, the KL divergence is ill-defined and cannot be used to train the models. When using DSM, the distribution after Gaussian convolution would always be *a.c.*, thus can be a valid training objective, see Zhang et al. [49] or Arjovsky et al. [1] for a detailed introduction.

## 2 Clean Model Identification

For a fixed $\sigma > 0$, DSM can only learn a 'noisy' data distribution $\tilde{q}_\theta(\tilde{x})$ even in the ideal case where the Fisher divergence is exactly minimized, since $\text{FD}(\tilde{p}_d||\tilde{q}_{\theta^*}) = 0 \to \tilde{p}_d = \tilde{q}_\theta \neq p_d$. In this case, the following theorem shows that there exists a 'clean' model that is implicitly defined that learns the true data distribution.

**Theorem 2.1** (Existence of the underlying clean model for optimal $\tilde{q}_\theta(\tilde{x})$). *When the Fisher divergence goes to 0, $\text{FD}(\tilde{p}_d||\tilde{q}_\theta) = 0 \to \tilde{p}_d = \tilde{q}_\theta$, there exists an **unique** underlying clean model $q(x)$ such that $\tilde{q}_\theta(\tilde{x}) = \int q(x)p(\tilde{x}|x)\,\mathrm{d}x$ and $q(x) = p_d(x)$.*

See Appendix A.1 for proof. This theorem shows that despite training an EBM on noisy data, there is an implicit model within it that can recover the true data distribution. Therefore, instead of annealing the noise $\sigma \to 0$ to recover the true data distribution, we will demonstrate how to directly sample from the implicitly-defined clean model given the noisy energy-based model in the next section.

We want to highlight that the 'perfect fit' assumption, i.e. achieving $\text{FD}(\tilde{p}_d||\tilde{q}_\theta) = 0$, may not hold for a complex data distribution $p_d$ or underpowered EBMs. Therefore, we provide general sufficient conditions for the existence of the clean model for an imperfect EBM in Appendix A.2.

### 2.1 Gibbs Sampling with Gaussian Moment Matching

Given a well-trained noisy energy-based model $\tilde{q}_\theta(\tilde{x}) = \tilde{p}_d(\tilde{x})$, the clean model has the form

$$q(x) = \int p(x|\tilde{x})\tilde{q}_\theta(\tilde{x})d\tilde{x}, \tag{5}$$

where the denoising distribution can be written as $p(x|\tilde{x}) \propto p(\tilde{x}|x)q(x)$. We notice that, since the noise distribution $p(\tilde{x}|x) = \mathcal{N}(x, \sigma^2 I)$ is known, a Gibbs sampling scheme can be constructed to sample from the underlying clean model if we know the denoising distribution $p(x|\tilde{x})$, with

$$\tilde{x}_{k-1} \sim p(\tilde{x}|x = x_{k-1}), \quad x_k \sim p(x|\tilde{x} = \tilde{x}_{k-1}), \tag{6}$$

where the initial sample $x_0 \sim p_0(x)$ can be drawn from a standard Gaussian $p_0(x) = \mathcal{N}(0, I)$. However, as the denoising distribution $p(x|\tilde{x})$ is usually intractable for complex $p_d$, we propose an analytical Gaussian moment matching approximation of $p(x|\tilde{x})$.

Denote the mean and covariance of $p(x|\tilde{x})$ as

$$\mu(\tilde{x}) = \langle x \rangle_{p(x|\tilde{x})}, \quad \Sigma(\tilde{x}) = \langle x^2 \rangle_{p(x|\tilde{x})} - \langle x \rangle^2_{p(x|\tilde{x})}. \tag{7}$$

The classic Gaussian moment matching method [24] specifies a Gaussian approximation $p(x|\tilde{x}) \approx \mathcal{N}(\mu(\tilde{x}), \Sigma(\tilde{x}))$, which matches the first and second moment of $p(x|\tilde{x})$. When $\tilde{q}_\theta = \tilde{p}_d$, the first mean of the denoised distribution has a well-known analytical form [39, 2, 9, 29]

$$\mu(\tilde{x}) = \tilde{x} + \sigma^2 s_{\tilde{q}_\theta}(\tilde{x}); \tag{8}$$

we include the derivation in Appendix A.3. Using this identity, we can rewrite Equation 4 as

$$\text{FD}(\tilde{p}_d||\tilde{q}_\theta) \doteq \frac{1}{2\sigma^4} \iint p(\tilde{x}|x)p_d(x)\,\|x - \mu(\tilde{x})\|^2_2\,\mathrm{d}\tilde{x}\,\mathrm{d}x, \tag{9}$$

where we can see that the Fisher divergence only depends on $\mu(\tilde{x})$. Since $\text{FD}(\tilde{p}_d||\tilde{q}_\theta) = 0 \Leftrightarrow q = p_d$ (Theorem 2.1), the function $\mu(\tilde{x})$ fully characterizes the distribution $q$. Therefore, $\mu(\tilde{x})$ and $p(\tilde{x}|x) = \mathcal{N}(x, \sigma^2 I)$ can provide sufficient information to determine $p(x|\tilde{x}) \propto q(x)p(\tilde{x}|x)$. As a consequence, the following theorem shows that the covariance function can also be analytically derived.

**Theorem 2.2** (Analytical Covariance Identity). *Given a clean model $q(x)$ such that $\int q(x)p(\tilde{x}|x)\,\mathrm{d}x = \tilde{q}_\theta(\tilde{x}) = \tilde{p}_d(\tilde{x})$ with $p(\tilde{x}|x) = \mathcal{N}(x, \sigma^2 I)$, the $\mu(\tilde{x})$ and $\Sigma(\tilde{x})$ of the $p(x|\tilde{x}) \propto q(x)p(\tilde{x}|x)$ has the following relations*

$$\Sigma(\tilde{x}) = \sigma^2 \nabla_{\tilde{x}}\mu(\tilde{x}) = \sigma^4 \nabla^2_{\tilde{x}} \log \tilde{q}_\theta(\tilde{x}) + \sigma^2 I. \tag{10}$$

See Appendix A.3 for proof. This analytical covariance identity can be seen as a high-dimensional generalization of the 2nd-order Tweedie's Formula [9, 29]. Therefore, the analytical full-covariance moment matching approximation can be written as

$$p(x|\tilde{x}) \approx \mathcal{N}(\tilde{x} + \sigma^2 \nabla_{\tilde{x}} \log \tilde{q}_\theta(\tilde{x}), \sigma^4 \nabla^2_{\tilde{x}} \log \tilde{q}_\theta(\tilde{x}) + \sigma^2 I). \tag{11}$$

We want to highlight that since the Gaussian moment matching is only an approximation of $p(x|\tilde{x})$, the sampling scheme in Equation 6 is a 'pseudo' Gibbs sampler unless the true $p(x|\tilde{x})$ is also a Gaussian distribution[3], which is not true for general non-Gaussian $p_d$. However, since $\mu(\tilde{x})$ and $p(\tilde{x}|x) = \mathcal{N}(x, \sigma^2 I)$ are already sufficient to specify $p(x|\tilde{x})$, it should be possible to derive expressions for higher-order moments which themselves involve only $\mu(\tilde{x})$ and $\sigma$; we leave this to future work. To our knowledge, the $\tilde{x}$-conditioned full covariance Gaussian moment matching approximation to $p(x|\tilde{x})$ has not been derived previously. In the next section, we briefly discuss the connections between our method and other related approaches.

## 2.2 Connection to Covariance Learning Approaches

Bengio et al. [4] proposes to approximate the true posterior $p(x|\tilde{x}) \propto p_d(x)p(\tilde{x}|x)$ with a variational distribution $q_\theta(x|\tilde{x})$. The parameter $\theta$ is then learned by minimizing the joint KL divergence

$$\mathrm{KL}(p(\tilde{x}|x)p_d(x)\|q_\theta(x|\tilde{x}))\tilde{p}_d(\tilde{x})) \doteq -\iint p_d(x)p(\tilde{x}|x)\log q_\theta(x|\tilde{x})\,\mathrm{d}\tilde{x}\,\mathrm{d}x, \qquad (12)$$

where $\tilde{p}_d(\tilde{x}) = \int p_d(x)p(\tilde{x}|x)\,\mathrm{d}x$. The joint KL divergence in Equation 12 encourages $q_\theta(x|\tilde{x})$ to match the moments of the true posterior $p(x|\tilde{x})$, and defines an upper bound of the marginal KL [48]

$$\mathrm{KL}(p(\tilde{x}|x)p_d(x)\|q_\theta(x|\tilde{x}))\tilde{p}_d(\tilde{x})) \geq \mathrm{KL}(p_d(x)\|q_\theta(x)), \qquad (13)$$

where the model is implicitly defined as the marginal of the joint $q_\theta(x) = \int q_\theta(x|\tilde{x})\tilde{p}_d(\tilde{x})$. When $q_\theta(x|\tilde{x})$ is a consistent estimator of $p(x|\tilde{x})$, this asymptotic distribution of the Gibbs sampling will converge to the true data distribution $p_d(x)$ [4]. For continuous data, the variational distribution is chosen as a Gaussian distribution $q_\theta(x|\tilde{x}) = \mathcal{N}(\mu_\theta(\tilde{x}), \Sigma_\theta(\tilde{x}))$, where the mean $\mu_\theta(\cdot)$ and the covariance $\Sigma_\theta(\cdot)$ are parameterized by neural networks. We note that the only difference between the KL and DSM objective (Equation 9) is that the KL objective additionally learns the covariance. We thus show that the optimal covariance under KL minimization is the proposed analytical covariance.

**Theorem 2.3** (Optimal Gaussian Approximation). *Let $p(\tilde{x}|x) = \mathcal{N}(0, \sigma^2 I)$ and assume Gaussian distribution $q_\theta(x|\tilde{x}) = \mathcal{N}(\mu_q(\tilde{x}), \Sigma_q(\tilde{x}))$, then the optimal $q^*$ such that*

$$q^* = \arg\min_q \mathrm{KL}(p(\tilde{x}|x)p_d(x)\|q(x|\tilde{x})\tilde{p}_d(\tilde{x})) \qquad (14)$$

*has the mean and covariance with the form*

$$\mu_q^*(\tilde{x}) = \langle x \rangle_{p(x|\tilde{x})}, \quad \Sigma_q^*(\tilde{x}) = \sigma^2 \nabla_{\tilde{x}} \mu_q^*(\tilde{x}), \qquad (15)$$

see Appendix A.4 for proof. Therefore, when the optimal mean function is learned $\mu_\theta(\tilde{x}) = \mu_q^*(\tilde{x})$, the optimal $\Sigma(\tilde{x})$ can be analytically derived, making the learning of $\Sigma_\theta(\tilde{x})$ redundant. In addition to the training inefficiency caused by more parameters, the amortized covariance network may suffer from poor generalization [50]. Moreover, the KL objective is also not well-defined for learning data distributions which lie on a low-dimensional manifold, e.g. MNIST, see Section 4 for a detailed discussion. In this case, the learned $\Sigma_\theta(\tilde{x})$ may be a degenerate matrix, making the Gaussian density function $q(x|\tilde{x})$ ill-defined [49] which impedes the training, see Figure 5 for an example.

Paper[23] proposes a higher-order score-matching loss to simultaneously learn both the first order score $\nabla_{\tilde{x}} \log \tilde{q}_\theta(\tilde{x})$ and the second order score $\nabla_{\tilde{x}}^2 \log \tilde{q}_\theta(\tilde{x})$. However, our findings indicate that the mean function $\mu(\tilde{x})$ (or the first order score $\nabla_{\tilde{x}} \log p(\tilde{x})$) already contains all the moment information of the underlying true distribution $p_d$, and the optimal moment can be derived using the mean function. Therefore, learning the second-order score is redundant and may lead to sub-optimal inference.

## 2.3 Connection to Analytic DDPM

The recent paper Bao et al. [2] considers a constrained variational family $q_\theta(x|\tilde{x}) = \mathcal{N}(\mu_\theta(\tilde{x}), \sigma_q^2 I)$ in the context of diffusion model and derive the optimal $\sigma_q^*$ as

$$\sigma_q^{*2} = \arg\min_{\sigma_q} \mathrm{KL}(p(\tilde{x}|x)p_d(x)\|q_\theta(x|\tilde{x}))\tilde{p}_d(\tilde{x})) = \frac{1}{d}\left\langle \mathrm{Tr}\left(\mathrm{Cov}_{q(x|\tilde{x})}[x]\right)\right\rangle_{\tilde{p}_d(\tilde{x})}, \qquad (16)$$

---

[3]For example, when $p_d(x) = \mathcal{N}(\mu_d, \sigma_d)$, the true posterior $p(x|\tilde{x}) \propto p_d(x)p(\tilde{x}|x)$ will be a Gaussian with mean $\mu(\tilde{x}) = (\sigma^2\mu_d + \sigma_d^2\tilde{x})/(\sigma^2 + \sigma_d^2)$ and variance $\Sigma(\tilde{x}) = \sigma_d^2\sigma^2/(\sigma^2 + \sigma_d^2)$, we can verify that $\Sigma(\tilde{x}) = \sigma^2\nabla_{\tilde{x}}\mu(\tilde{x})$.

which can also be rewritten using the score function

$$\sigma_q^{*2} = \sigma^2 - \sigma^4/d \left\langle \|s_{q_\theta}(\tilde{x})\|_2^2 \right\rangle_{\tilde{p}_d(\tilde{x})}. \tag{17}$$

In Appendix B, we provide a detailed derivation to show how this approximation can be linked to our method using the Fisher information identity [10]. This approximation has two potential limitations: first, compared to full covariance moment matching, the assumed isotropic covariance structure may be insufficiently flexible to capture the true posterior; second, the covariance is independent of $\tilde{x}$.

The second assumption only holds when $\mu(\tilde{x})$ is a linear function of $\tilde{x}^4$ (e.g. when $p_d(x)$ is Gaussian) and does not hold for other non-Gaussian $p_d(x)$. Therefore, our $\tilde{x}$-dependent full-covariance approximation offers a more versatile approximation family, which ultimately results in a more precise estimation. However, in certain applications such as accelerating the sampling procedure of a diffusion model [2], it is advantageous to use a $\tilde{x}$-independent isotropic covariance due to its inexpensive estimation. On the other hand, our $\tilde{x}$-dependent covariance necessitates the computation of the Hessian for each $\tilde{x}$, making it inefficient for high-dimensional data. In Section 3, we will explore approaches to mitigate this limitation.

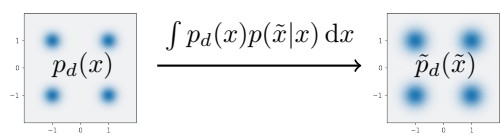

(a) Gaussian mixtures visualizations

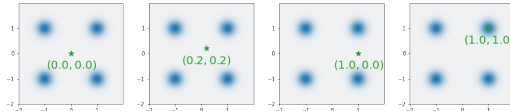

(b) Density of $p(x|\tilde{x}')$ (blue) and four $\tilde{x}'$ points (green)

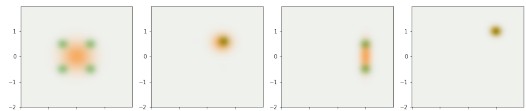

(c) Analytical full covariance moment matching

### 2.4 Posterior Approximation Comparison

We now consider a toy example to compare the three denoising posterior approximations discussed above. Let $p_d(x)$ be a Mixture of Gaussians (MoG) $p_d(x) = \frac{1}{4}\sum_{k=1}^{k=4} g_k(x)$ whose components $g_{[1:4]}$ are 2D Gaussians with means $[-1,-1], [-1,1], [1,1], [1,-1]$ and isotropic covariance $\sigma_g^2 I$ with $\sigma_g = 0.2$. The noise distribution is $p(\tilde{x}|x) = \mathcal{N}(x, \sigma^2 I)$ with $\sigma = 0.2$, so $\tilde{p}(\tilde{x}) = \int p(x)p(\tilde{x}|x)\,\mathrm{d}x$ is an MoG with the same component means and diagonal covariance $(\sigma_g^2 + \sigma^2)I$; see Figure 1a for a visualization. In this case the true posterior $p(x|\tilde{x})$ does not allow a tractable form. Fortunately, given a noisy sample $\tilde{x}'$ and an evaluation point $x'$, we can evaluate the true density $p(x|\tilde{x}')$ using Bayes rule: $p(x'|\tilde{x} = \tilde{x}') = p(\tilde{x}'|x = x')p_d(x')/\tilde{p}_d(\tilde{x}')$. Figure 1 shows the true posteriors given four different $\tilde{x}'$ where we use grid data in $x$-space to visualize the density.

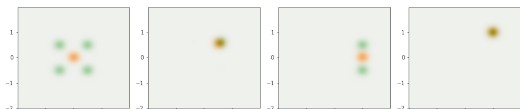

(d) Analytical isotropic covariance moment matching

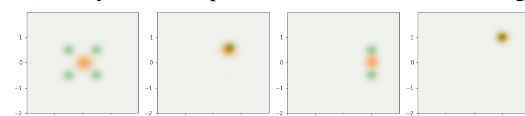

(e) Learned diagonal covariance

Figure 1: Figure (a) shows the clean data distribution $p_d(x)$ and the corresponding noisy distribution $\tilde{p}_d(\tilde{x})$. Figure (b) shows 4 conditioned samples in the noisy space. Figures (c, d, e) visualize the true posterior $p(x|\tilde{x})$ (green) and three posterior approximations (orange). We find that only the proposed $\tilde{x}$-dependent analytical full-covariance moment matching can capture the variance of the true posterior, whereas the other two methods underestimate the variance.

To train the model, we sample 10,000 data points from $p_d$ as our training data. For the KL-trained Gibbs sampler described in Section 2.2, we use a network with 3 hidden layers with 400 hidden units, Swish activation [28] and output size 4 to generate both mean and log standard deviation of the Gaussian approximation. For the moment-matching Gibbs sampler (including both full and isotropic covariance), we use the same network architecture but with output size 1 to get the scalar energy and DSM as the training objective. Both networks are trained with batch size 100 and Adam [19] optimizer with learning rate $1 \times 10^{-4}$ for 100 epochs. For the $\tilde{x}$-independent isotropic covariance, we use the Monte Carlo approximation to estimate the variance [2] with 10000 samples from $\tilde{p}_d(\tilde{x})$.

---

[4]Since when $\mu(\tilde{x})$ is a linear function of $\tilde{x}$, using Theorem 2.2, we have $\Sigma(\tilde{x}) = \sigma^2 \nabla_{\tilde{x}}\mu(\tilde{x})$ will not depend on $\tilde{x}$, see also footnote 1 for an example.

Figure 1 visualizes the approximations to the denoising posterior $p(x|\tilde{x})$ estimated by each of the three methods described in the previous sections. We surprisingly find that although the KL objective in Equation 12 encourages $q_\theta(x|\tilde{x})$ to match the moments of

Table 1: MMD evaluations of a single chain

| Data | Learn diag. | Analytic iso. | Analytic full |
|------|-------------|---------------|---------------|
| MoGs | $0.929 \pm 0.343$ | $0.724 \pm 0.361$ | $\mathbf{0.305} \pm 0.141$ |
| Rings | $0.364 \pm 0.044$ | $0.006 \pm 0.002$ | $\mathbf{0.005} \pm 0.001$ |
| Roll | $0.053 \pm 0.011$ | $0.030 \pm 0.001$ | $\mathbf{0.016} \pm 0.002$ |

$p(x|\tilde{x})$, the learned covariance in Figure 1e still underestimates the variance of the posterior. This shows the redundancy of covariance learning can degrade the variational approximation performance. Additionally, the $\tilde{x}$-independent covariance fails to account for the relative positions of $\tilde{x}'$ and lacks the ability to predict the posterior's elliptical shape due to its isotropic nature. In contrast, our $\tilde{x}$-dependent full covariance approximation overcomes these limitations, enabling more accurate predictions that capture the intricate geometry of the posterior distribution.

We then use the estimated posterior to conduct (pseudo) Gibbs sampling to generate samples[5]. Specifically, we initialize the first sample $x_0 \sim \mathcal{N}(0, 0.1)$ and run one Markov Chain with 10,000 time steps to generate 10,000 samples. In addition to the mixture of Gaussian datasets, we also train and generate samples from the 2D Swiss roll and two-ring datasets. For numerical evaluation, we calculate the Maximum Mean Discrepancy (MMD) [12] between 10k samples generated by a single-chain Gibbs sampler and 10k samples from the training dataset respectively. The kernel insides MMD is a sum over 5 Gaussian kernels with bandwidth ranging over $[2^{-2}, 2^{-1}, 2^0, 2^1, 2^2]$. The MMD results (including both mean and std) are calculated using 5 random seeds. We find that Gibbs sampling with the

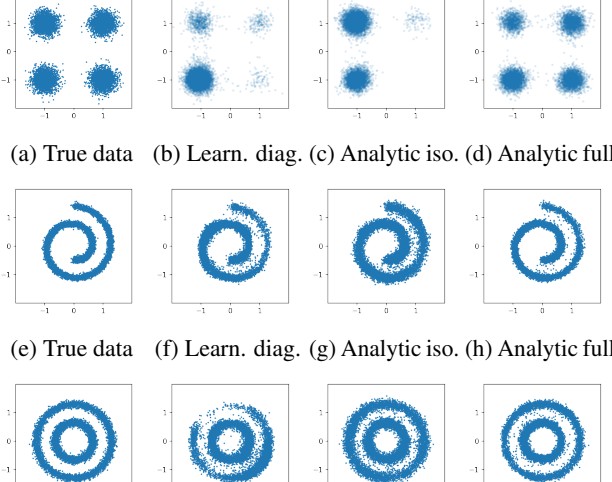

(a) True data  (b) Learn. diag.  (c) Analytic iso.  (d) Analytic full

(e) True data  (f) Learn. diag.  (g) Analytic iso.  (h) Analytic full

(i) True data  (j) Learn. diag.  (k) Analytic iso.  (l) Analytic full

Figure 2: Samples from a single chain Gibbs sampling

proposed analytical full covariance achieves the best results; numerical results are in Table 1, with a visual comparison in Figure 2.

## 3 Scalable Implementations for Image Data

**Scalable Diagonal Hessian Approximation** As we discussed in Section 2.3, the proposed full covariance Gaussian approximation in Equation 10 requires calculating an $D \times D$ Hessian $\nabla_{\tilde{x}}^2 \log \tilde{q}(\tilde{x})$ for each $\tilde{x}$ with size $D$, which brings both memory and computation difficulties for high-dimensional data. A naive diagonal Hessian method (only using the diagonal entries in the Hessian) will address the memory bottleneck but still needs $D$ times backward passes for the exact computation of the diagonal term [22]. In this paper, we use the following diagonal Hessian approximation [3],

$$\text{Diag}(H) \approx 1/S \sum_{s=1}^{S} v_s \odot H v_s, \tag{18}$$

where $v_s \sim p(v)$ is a Rademacher random variable with entries $\pm 1$ and $\odot$ denotes the element-wise product[6]. This estimator will converge to the exact Hessian diagonals when $S \to \infty$ [3]. The computation for each $v_s$ can be computed by two forward-backward passes. It is worth emphasizing that our $\tilde{x}$-dependent diagonal moment matching approach provides a comparable level of flexibility to the variational method proposed in [4] while eliminating the need for additional training of the

---

[5]The code of the experiments can be found in `https://github.com/zmtomorrow/MMDGS_NeurIPS`.

[6]This estimation should be distinguished from the Hutchinson's Trace estimation [15]: $\text{Tr}(H) \approx \frac{1}{S} \sum_{s=1}^{S} v_s^T H v_s$ where a dot-product is used between $v_s$ and $H v_s$.

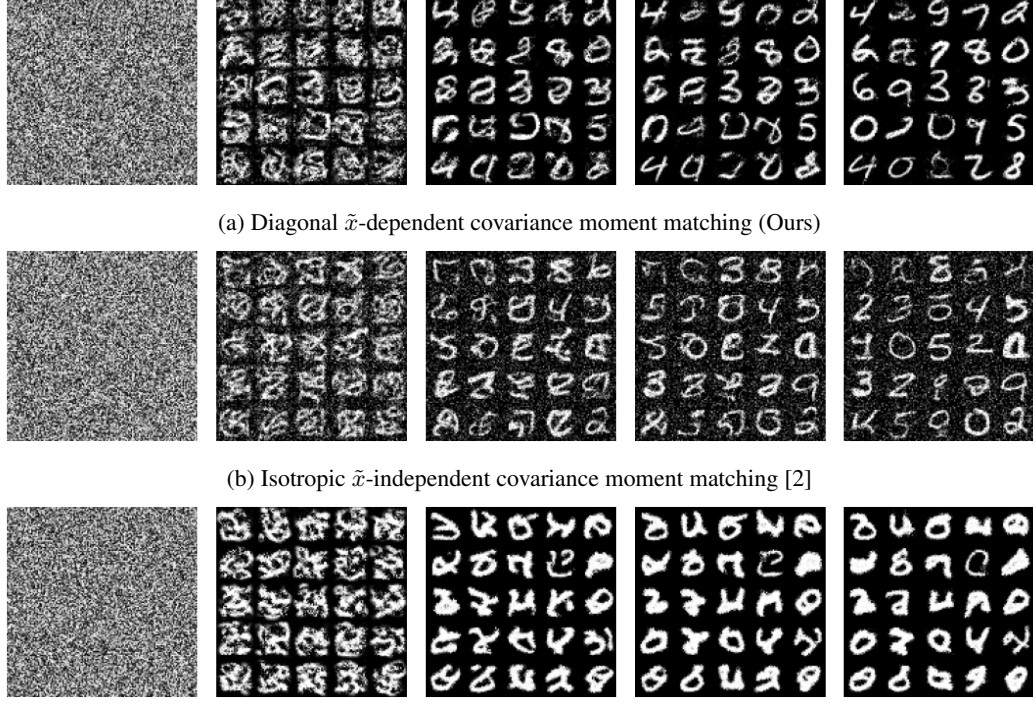

(a) Diagonal $\tilde{x}$-dependent covariance moment matching (Ours)

(b) Isotropic $\tilde{x}$-independent covariance moment matching [2]

(c) Diagonal covariance learned by KL minimization [4]

Figure 3: Figures (a,b,c) show the MNIST experiment comparisons, where we compare samples generated by pseudo-Gibbs sampling with three different $q(x|\tilde{x})$. We plot samples from 25 independent Markov Chains with $t \in \{0, 1, 5, 10, 20\}$ time steps. We can find the samples generated by the proposed analytical covariance moment matching with diagonal approximation achieved the best sample quality.

diagonal covariance. Furthermore, our method remains more flexible than the isotropic $\tilde{x}$-independent moment matching method proposed by [2].

**Energy or Score Parameterization** For the full-covariance moment matching in Equation 10, we require $\nabla_{\tilde{x}}^2 \log \tilde{q}_\theta(\tilde{x})$ to be symmetric to obtain a valid Gaussian approximation. However, if we learn the score function $\nabla_{\tilde{x}} \log p(\tilde{x}) = s_\theta(\tilde{x})$ using a network $s_\theta(\cdot) : \mathbb{R}^D \to \mathbb{R}^D$, its Jacobian is not guaranteed to be symmetric. In this case, we follow [34] and directly parameterize the density function $\tilde{q}_\theta(\cdot)$ with a neural network $f_\theta(\cdot) : \mathbb{R}^D \to \mathbb{R}$ and let the score function $\nabla_{\tilde{x}} \log \tilde{q}_\theta(\tilde{x}) = -\nabla_x f_\theta(\tilde{x})$. This can be obtained by AutoDiff packages like PyTorch [27], and this parameterization guarantees $\nabla_{\tilde{x}}^2 f_\theta(\tilde{x})$ to be symmetric. We also notice that when using the diagonal Hessian approximation (Equation 18), we only need entries in $\mathrm{Diag}(H)$ to be positive in order to obtain a valid Gaussian approximation. In this case, the score parameterization remains applicable and offers more efficient training compared to the energy parameterization. Therefore, the combination of full/diagonal covariance and energy/score parameterization provides a tradeoff between flexibility and inference speed, allowing for a flexible approach while maintaining computational efficiency during training.

## 4 Image Generation with a Single Noise Level

We then apply the proposed method to model the grey-scale MNIST [21] dataset. We use the standard U-Net architecture [35, 31] with a single fixed noise level $\sigma = 0.5$; the effect of varying $\sigma$ is explored in Appendix C.1. For the KL training objective, the output channel size is 2 to generate both mean and log-std at the same time. For DSM training, we take the sum of the U-Net output to obtain the scalar energy evaluation which also relates to the product-of-experts model described in [34]. We train both networks for 300 epochs with learning rate $1 \times 10^{-4}$ and batch-size 100.

As discussed in Section 1.1, the KL divergence is not well-defined for manifold data distributions. This limitation becomes evident when working with MNIST, where the presence of constant black pixels in the boundary areas leads to a rapid decrease towards 0 in the variance of $q_\theta(x|\tilde{x})$ during training. Consequently, the likelihood value $\log q_\theta(x|\tilde{x})$ tends to approach infinity, resulting in unstable training. In contrast, the DSM objective is well-defined for manifold data, providing



(a) $\tilde{x}$-dep diag (Ours) (b) $\tilde{x}$-dep diag. (KL)    (c) $\tilde{x}$-ind iso.

Figure 4: Figures (a,b,c) visualize the covariance approximations $q(x|\tilde{x} = x + \sigma\epsilon), \epsilon \sim \mathcal{N}(0, \sigma^2 I)$ on 25 $\tilde{x}$ samples. We use a sigmoid function to map the real value noise into grayscale pixels for the visualization.

a stable training process even in the presence of such boundary effects. Figure 5 provides a visual comparison of the two training procedures, demonstrating the improved stability and effectiveness of the DSM objective in handling manifold data distributions.

For the sample generation process, calculating the full-covariance Gaussian posterior becomes challenging. We therefore apply the scalable diagonal Hessian approximation described in Section 3 to approximate the diagonal Gaussian covariance of $p(x|\tilde{x})$. We find that the estimated diagonal Hessian occasionally contains small negative values due to approximation error; we, therefore, use the $\max(\cdot, \epsilon)$ function with $\epsilon > 0$ to ensure the pos-

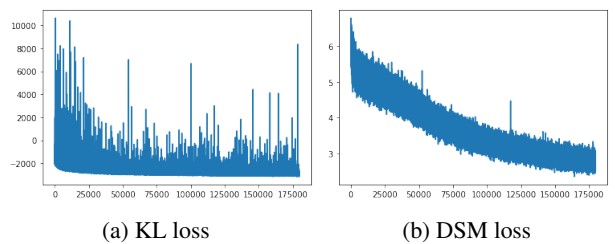

(a) KL loss                (b) DSM loss

Figure 5: Training loss comparison of two objectives. We plot the training loss every iteration during a total 300 epochs.

itivity of the diagonal covariance. The $\tilde{x}$-independent isotropic covariance and the proposed $\tilde{x}$-dependent diagonal covariance share the same mean function.

We first visualize the covariance estimated by three different methods in Figure 4. We use 100 Rademacher samples in estimating the diagonal Hessian (Equation 18) and 50,000 samples in estimating the isotropic variance (Equation 17). We find that both $\tilde{x}$-dependent diagonal covariance approximations can capture the posterior structure whereas the isotropic

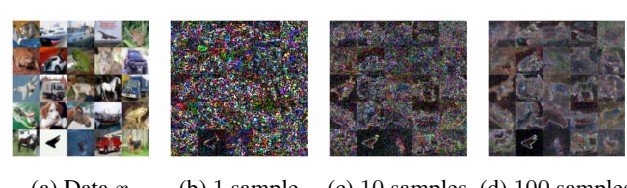

(a) Data $x$     (b) 1 sample    (c) 10 samples  (d) 100 samples

Figure 6: Visualizations of the diagonal covariance $q(x|\tilde{x} = x + \sigma\epsilon)$ with different number of Rademacher samples.

$\tilde{x}$-independent covariance is just Gaussian noise since the variance is shared between different digit and pixel locations. In Figure 3, we plot the sample comparison for three methods.

Since the isotropic covariance has the same variance in each dimension, the generated samples in Figure 3b contain white noise in the black background, whereas the proposed full-covariance sampler

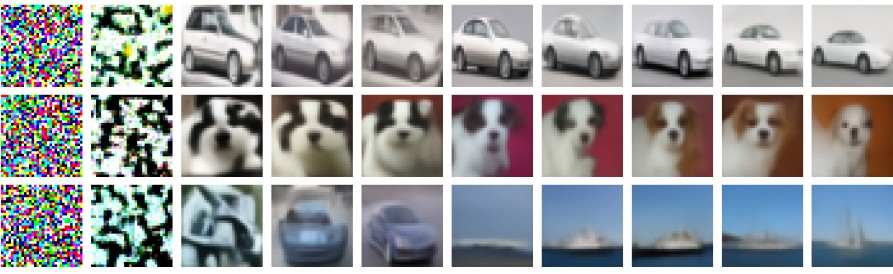

Figure 7: Samples from three Markov chains. We plot the samples every 10 Gibbs steps.

| **Algorithm 1** Sampling with Langevin Dynamics | **Algorithm 2** Sampling with the proposed pseudo Gibbs Sampling |
|---|---|
| **Require:** $\{\sigma_t\}_{t=0}^T, \delta, K$
  Initialize $x_T^0$
  **for** $t \leftarrow T$ to 1 **do**
    $\alpha_t \leftarrow \delta\sigma_t^2/\sigma_0^2$
    **for** $k \leftarrow 1$ to $K$ **do**
      Draw $z^k \sim \mathcal{N}(0, I)$
      $x_t^k \leftarrow x_t^{k-1} + \alpha_t s_\theta(x_t^{k-1}, \sigma_t) + \sqrt{2\alpha_t}z^k$
    **end for**
    $x_{t-1}^0 \leftarrow x_t^K$
  **end for**
  return $x_0^0 + \sigma_0^2 s_\theta(x_0^0, \sigma_0)$ | **Require:** $\{\sigma_t\}_{t=0}^T, K$
  Initialize $x_T^0$
  **for** $t \leftarrow T$ to 1 **do**
    **for** $k \leftarrow 1$ to $K$ **do**
      Draw $x_{t+1}^k \sim p(x_{t+1}|x_t = x_t^k)$
      Draw $x_t^k \sim q_\theta(x_t|x_{t+1} = x_{t+1}^k)$
    **end for**
    $x_{t-1}^0 \leftarrow x_t^K$
  **end for**
  return $x_0^0 + \sigma_0^2 s_\theta(x_0^0, \sigma_0)$ |

can generate a clean black background in Figure 3a. On the other hand, the samples generated by the KL-trained Gibbs sampler (Figure 3c) have worse sample quality due to the unstable training.

We then apply the same method to model the more complicated CIFAR 10 [20] dataset. We use the same U-Net structure as used in [36] and directly parameterize the score function rather than the energy function to speed up the training. The noise level is fixed at 0.3. We train the model using Adam optimizer with learning rate $1 \times 10^{-4}$ and batch size 100 for 1000 epochs. We visualize the denoising posterior diagonal covariance in Figure 6 when using different numbers of Rademacher samples (Equation 18). We observe that better covariance estimation can be obtained by increasing the number of samples. To balance efficiency and accuracy, we use a sample number of 10 in the subsequent Gibbs sampling stage. Figure 7 shows three independent Markov chains with the samples plotted every 10 Gibbs steps, which demonstrates that sharp images can be generated with even one fixed level of noise.

**Limitation:** In the CIFAR experiment, we observe a mode collapse phenomenon when running multiple independent Markov chains for a longer time. This phenomenon is likely due to the small noise level $\sigma = 0.3$, which prevents the sampler from exploring the full space, as commonly found with MCMC methods [30]. This effect is visually represented in Figure 8, where we assess the Fréchet Inception Distance (FID) values for 50,000 images sampled with varying numbers of Gibbs steps. Notably, the FID increases beyond 40 Gibbs steps, and visual evidence of mode collapse is observed (Figure 12). In the ensuing section, we will demonstrate the application of our method to settings with multiple noise levels, an approach that may help mitigate the issue of mode collapse.

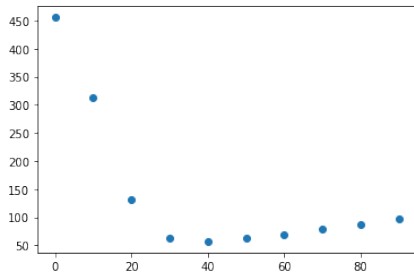

Figure 8: FID evaluation with Increased Gibbs Steps. We can find the FID increases after 40 Gibbs steps.

## 5 Image Generation Using Multiple Noise Levels

The success of diffusion models and lessons from prior work on score-based generative models point to the importance of using multiple noise levels [14, 35] when modelling data with complex multi-modal distributions. Intuitively, by learning to denoise data at a range of noise levels, a single network can learn both the fine and global structure of the distribution, which in turn allows for more effective sampling algorithms capable of efficiently exploring diverse modes [35]. We therefore propose to adapt the denoising Gibbs sampling procedure to sample from distributions corrupted with multiple noise levels. For this purpose, we use a noise-conditioned score network trained by Song and Ermon [36], who generated high-quality samples using a procedure inspired by annealed Langevin dynamics [43]. This procedure involves generating samples from a sequence of distributions $p_T(x_T), ..., p_0(x_0)$, corrupted by progressively decreasing levels of Gaussian noise (parameterized via standard deviations $\sigma_t$, with $\sigma_T >, \cdots, > \sigma_0$). At a given step $t$ in the sequence, Langevin dynamics is used to sample from the corresponding noised distribution $p_t(x_t)$, using the

score network $s_\theta(x_t, \sigma_t)$ to approximate the gradient of the noised distribution. The outputs of this Langevin dynamics run are then used to initialize the same procedure at the next noise level, leading the sampling procedure to converge gradually towards the data distribution as the noise level tends to zero (i.e. $p_0(x_0) \approx p_d(x), \sigma_0 \approx 0$). The algorithm of the annealed Langevin dynamics with multi-level noise used in [35, 36] is summarized in Algorithm 1.

We show that the proposed Gibbs sampling scheme can be directly applied to a pre-trained score-based generative model as a drop-in replacement for Langevin dynamics MCMC in the generation stage. At each noise level, we use samples $x_{t+1}$ from the previous noise level to initialise a Gibbs sampling chain targeting the marginal distribution at the current noise level $p_t(x_t)$, which now plays the role of the 'clean' distribution in Equation 5. Therefore, the noisy distribution at time step $t$ is a Gaussian $p(x_{t+1}|x_t) = \mathcal{N}(x_t, \sigma_{t+1}^2 - \sigma_t^2)$. The optimal denoising distribution $q(x_t|x_{t+1})$ is thus a function of the level of noise at step $t$ relative to the level of noise at the previous step $t + 1$. For generation efficiency, we employ 3 Gibbs steps at each noise level, using 3 Rademacher samples to approximate the diagonal Hessian. The sampling procedure is summarized in Algorithm 2. In Figure 9 and 10, we visualize the samples from models that are trained on CIFAR10 and CelebA separately. Further experimental details can be found in Appendix C.2.

For direct comparison with the results of [36] on CIFAR10, we retain the same schedule of noise levels used to generate samples with Langevin dynamics. We generate 50000 samples using this approach and report FID and Inception scores in Table 2. Our multi-level Gibbs sampling scheme produces samples of equivalent quality to the multi-level Langevin

Table 2: CIFAR10 Inception and FID Scores

| Model | Inception | FID |
|---|---|---|
| NCSNv2 (Langevin [36]) | $\mathbf{8.40 \pm 0.07}$ | 10.87 |
| NCSNv2 (Gibbs, Ours) | $8.28 \pm 0.07$ | **10.75** |
| DDPM [14] | $9.46 \pm 0.11$ | 3.17 |
| NCSN++ [39] | 9.89 | 2.20 |

dynamics of [36], confirming its applicability to complex natural image data. The FID is also notably superior to that of the single-noise level Gibbs sampling, and the samples exhibit significant visual diversity (Figure 9). This underlines the importance of employing multi-level noise in our approach. Recent advances in sampling strategies for score-based models leveraging the framework of stochastic differential equations [39] have led to significant further improvements in generation quality as shown in Table 2; we leave the exploration of possible applications of our method to this framework to future work.

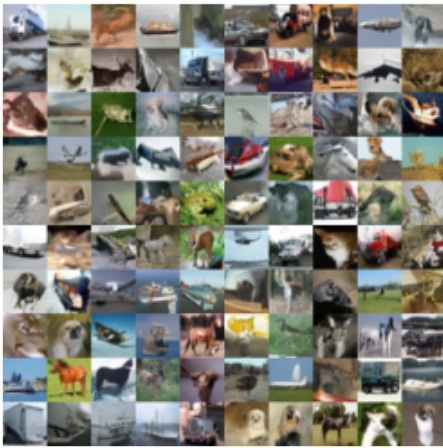

Figure 9: CIFAR 10 Samples

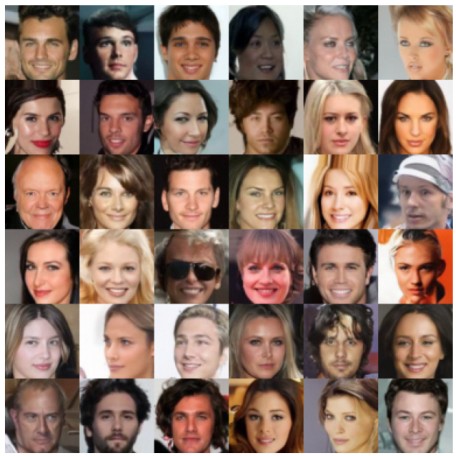

Figure 10: CelebA Samples

## 6   Conclusion

This paper focuses on addressing the inconsistency problem in training energy-based models (EBMs) using denoising score matching. Specifically, we identify the presence of an underlying clean model within a 'noisy' EBM and propose an efficient sampling scheme for the clean model. We demonstrate how this method can be effectively applied to high-dimensional data and showcase image generation results in both single and multi-level noise settings. More broadly, we hope our more accurate denoising posterior opens new avenues for future work on score-based methods in machine learning.

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

# A Proof and Derivations

## A.1 Proof of Theorem 2.1

The existence is straightforward, since $\mathrm{FD}(\tilde{p}_d||\tilde{q}_{\theta^*}) = 0 \rightarrow \tilde{p}_d = \tilde{q}_{\theta^*}$, we can simply let $q(x) = p_d(x)$, which makes $\int q(x)p(\tilde{x}|x)\,\mathrm{d}x = \int p_d(x)p(\tilde{x}|x)\,\mathrm{d}x = \tilde{p}_d$. To show the uniqueness, we denote density $k(\epsilon) = \mathcal{N}(0, \sigma^2 I)$, so $\tilde{q}_\theta(\tilde{x})$ and $\tilde{p}_d(x)$ can be written as convolutions

$$\tilde{q}_\theta(\tilde{x}) = q * k, \quad \tilde{p}_d(\tilde{x}) = p_d * k, \tag{19}$$

we then have

$$\tilde{p}_d = \tilde{q}_\theta \Leftrightarrow q * k = p_d * k \Leftrightarrow \mathcal{F}(q)\mathcal{F}(k) = \mathcal{F}(p_d)\mathcal{F}(k), \tag{20}$$

where $\mathcal{F}$ denotes the Fourier transform. Since the Fourier transform of a Gaussian is also a Gaussian, so $\mathcal{F}(k) > 0$ everywhere, we have

$$\tilde{p}_d = \tilde{q}_{\theta^*} \Leftrightarrow \mathcal{F}(q)\cancel{\mathcal{F}(k)} = \mathcal{F}(p_d)\cancel{\mathcal{F}(k)} \Leftrightarrow \mathcal{F}(q) = \mathcal{F}(p_d) \Leftrightarrow q = p_d. \tag{21}$$

Therefore, $q = p_d$ is the unique distribution that makes $\tilde{p}_d = \tilde{q}_\theta$. This technique has also been used to construct spread KL divergence (we denote as $\widetilde{\mathrm{KL}}$) [49], which is defined as $\widetilde{\mathrm{KL}}(p_d||q_\theta) \equiv \mathrm{KL}(p_d * k||q_\theta * k)$ where $k(\epsilon) = N(0, \sigma^2 I)$, to train implicit model $q_\theta$. Different from the DSM situation, when $\widetilde{\mathrm{KL}}(p_d||q_\theta) = 0$, the underlying model $q_\theta = p_d$ is directly available, whereas the EBM $\tilde{q}_\theta$ trained by DSM learns to be the noisy distribution $\tilde{q}_\theta = p_d * k$.

## A.2 General Conditions Characterising the Existence of the Clean Model

In the previous section, we assume for a flexible neural network parameterized $f_\theta$, the energy-based model $\tilde{q}_\theta(\tilde{x}) = \exp(-f(\tilde{x}))/Z(\theta)$ trained by Equation 4 can recover the target noisy data distribution $\tilde{q}_{\theta^*} = \tilde{p}_d$ so there exists an underlying model $q$ such that $\tilde{q}_{\theta^*} = q * k$ and $q = p_d$. This assumption is commonly used in the literature on score-based methods. For example, in the score-based diffusion models literature [35, 14, 2], for any data $x \in \mathbb{R}^D$, the score function $\nabla_{\tilde{x}} \log \tilde{q}_\theta(\tilde{x})$ is usually parameterized by a neural network $\mathrm{NN}_\theta(\cdot) : \mathbb{R}^D \rightarrow \mathbb{R}^D$. However, this parameterization cannot guarantee $\mathrm{NN}_\theta(\tilde{x})$ is a conservative vector field, or in other words, there doesn't exist a distribution $\tilde{q}_\theta(\tilde{x})$ such that $\nabla_{\tilde{x}}\tilde{q}_\theta(\tilde{x}) = \nabla_{\tilde{x}} \log \tilde{q}_\theta(\tilde{x})$ and $\nabla_{\tilde{x}}^2 \log \tilde{q}(\tilde{x})$ is symmetric [32, 33]. Therefore, perfect score estimation $\nabla_{\tilde{x}} \log \tilde{p}_d(\tilde{x}) = \nabla_{\tilde{x}} \log \tilde{q}_\theta(\tilde{x})$ is implicitly assumed to allow an EBM interpretation.

However, the underlying clean model doesn't always exist for imperfect model $\tilde{q}_\theta \neq \tilde{p}_d$. We here provide the sufficient and necessary conditions which guarantee the existence of the underlying clean model.

**Theorem A.1** (Necessary and Sufficient conditions for the existence of the underlying clean model.)**.** *For a model $\tilde{q}_\theta$ with the convolutional noise distribution $k(\epsilon) = \mathcal{N}(0, \sigma^2 I)$, there exists an underlying model $q$ such that $q * k = \tilde{q}$ if and only if $\mathcal{F}(\tilde{q}_\theta)/\mathcal{F}(k)$ is positive semi-definite [7]. Additionally, the underlying distribution $q$ can be written as*

$$q = \mathcal{F}^{-1}(\mathcal{F}(\tilde{q}_\theta)/\mathcal{F}(k)), \tag{22}$$

where $\mathcal{F}^{-1}$ is the inverse Fourier transform. This theorem is a straightforward corollary of Bochner's Theorem [8]. However, for the energy model $\tilde{q}_\theta(\tilde{x}) \propto \exp(-f_\theta(\tilde{x}))$, it's difficult to design a functioning family of $f$ that satisfies the positive semi-definite condition and have the tractable score function at the same time [9]. We thus leave the design of better energy function parameterizations as a promising future direction.

---

[7] A continuous function $f : \mathbb{R}^d \rightarrow \mathbb{C}$ is positive semi-definite if for all $n \in \mathbb{N}$, all sets of pairwise distinct centers $X = \{x_1, ..., x_N\} \in \mathbb{R}^d$ and all $\alpha \in \mathbb{C}^N$, $\sum_{i=1}^N \sum_{j=1}^N \alpha_i \overline{\alpha_j} f(x_i - x_j) \geq 0$, see [44, Definition 6.1]

[8] Bochner's Theorem [44, Theorem 6.6]: A continuous function $f : \mathbb{R}^d \rightarrow \mathbb{C}$ is positive semi-definite if and only if it is the Fourier transform of a finite non-negative Borel measure on $\mathbb{R}^d$.

[9] For example, one can define a noisy energy-based model $\tilde{q}_\theta = \exp(-f_\theta(\tilde{x}))/Z(\theta)$ with $-f_\theta(\tilde{x}) = \log \int (-g_\theta(x) - 1/\sigma^2||\tilde{x} - x||_2^2)\,\mathrm{d}x$, which always allows an underlying clean energy-based model $q_\theta(x) = \exp(-g_\theta(x))/Z(\theta)$ such that $\tilde{q}_\theta(\tilde{x}) = q_\theta(x) * k$ with $k(\epsilon) = \mathcal{N}(0, \sigma^2 I)$. However, the score function $\nabla_{\tilde{x}} \log \tilde{q}(\tilde{x}) = -\nabla_{\tilde{x}} f_\theta(\tilde{x})$ is intractable in this case.

### A.3 Proof of Theorem 2.2

**Derivation of the Mean Identity**

We let $\tilde{q}_\theta(\tilde{x}) = \int k(\tilde{x}|x)q_\theta(x)\,\mathrm{d}\tilde{x}$, where $k(\tilde{x}|x) = \mathcal{N}(x, \sigma^2 I)$, we have

$$\nabla_{\tilde{x}} \log \tilde{q}_\theta(\tilde{x}) = \frac{\nabla_{\tilde{x}}\tilde{q}_\theta(\tilde{x})}{\tilde{q}_\theta(\tilde{x})} = \frac{\int \nabla_{\tilde{x}}k(\tilde{x}|x)q_\theta(x)\,\mathrm{d}x}{\tilde{q}_\theta(\tilde{x})}$$

$$= -\frac{1}{\sigma^2}\int\left((\tilde{x}-x)\frac{k(\tilde{x}|x)q_\theta(x)}{\tilde{q}_\theta(\tilde{x})}\right)\mathrm{d}x$$

$$\implies \sigma^2\nabla_{\tilde{x}}\log\tilde{q}_\theta(\tilde{x}) + \tilde{x} = \int x\frac{k(\tilde{x}|x)q_\theta(x)}{\tilde{q}_\theta(\tilde{x})}\,\mathrm{d}x = \langle x\rangle_{q_\theta(x|\tilde{x})}$$

where we define the model denoising posterior using Bayes rule $q_\theta(x|\tilde{x}) \equiv k(\tilde{x}|x)q_\theta(x)/\tilde{q}_\theta(\tilde{x})$. The second equality is due to the following Gaussian distribution property

$$\nabla_{\tilde{x}}k(\tilde{x}|x) = \frac{1}{\sqrt{2\pi\sigma^2}}\nabla_{\tilde{x}}e^{\frac{-(\tilde{x}-x)^2}{2\sigma^2}} = -\frac{\tilde{x}-x}{\sigma^2}\frac{1}{\sqrt{2\pi\sigma^2}}e^{\frac{-(\tilde{x}-x)^2}{2\sigma^2}} = -\frac{\tilde{x}-x}{\sigma^2}k(\tilde{x}|x). \quad (23)$$

**Derivations of the Analytical Full Covariance Identity**

We have derived the mean identity

$$\mu_q(\tilde{x}) \equiv \langle x\rangle_{q_\theta(x|\tilde{x})} = \sigma^2\nabla_{\tilde{x}}\log\tilde{q}_\theta(\tilde{x}) + \tilde{x}. \quad (24)$$

Taking the gradient over $x$ in both side and scaling with $\sigma^2$, we have

$$\sigma^2\nabla_{\tilde{x}}\mu_q(\tilde{x}) = \sigma^4\nabla^2_{\tilde{x}}\log\tilde{q}_\theta(\tilde{x}) + \sigma^2 I. \quad (25)$$

We can also expand the hessian of the $\log\tilde{q}_\theta(\tilde{x})$:

$$\nabla^2_{\tilde{x}}\log\tilde{q}_\theta(\tilde{x}) = -\frac{1}{\sigma^2}\int\nabla_{\tilde{x}}\left((\tilde{x}-x)\frac{k(\tilde{x}|x)q_\theta(x)}{\tilde{q}_\theta(\tilde{x})}\right)\mathrm{d}x$$

$$= -\frac{1}{\sigma^2}\int\frac{k(\tilde{x}|x)q_\theta(x)}{\tilde{q}_\theta(\tilde{x})}\,\mathrm{d}x + \frac{1}{\sigma^2}\int(\tilde{x}-x)\frac{\nabla_{\tilde{x}}k(\tilde{x}|x)\tilde{q}_\theta(\tilde{x})q_\theta(x) - \nabla_{\tilde{x}}\tilde{q}_\theta(\tilde{x})k(\tilde{x}|x)q_\theta(x)}{\tilde{q}_\theta^2(\tilde{x})}\,\mathrm{d}x$$

$$\implies \sigma^2\nabla^2_{\tilde{x}}\log\tilde{q}_\theta(\tilde{x}) + 1 = \int(\tilde{x}-x)\frac{\nabla_{\tilde{x}}k(\tilde{x}|x)q_\theta(x) - \nabla_{\tilde{x}}\log\tilde{q}_\theta(\tilde{x})k(\tilde{x}|x)q_\theta(x)}{\tilde{q}_\theta(\tilde{x})}\,\mathrm{d}x$$

$$= \int(\tilde{x}-x)\frac{-\frac{1}{\sigma^2}(\tilde{x}-x)k(\tilde{x}|x)q_\theta(x) + \frac{1}{\sigma^2}(\tilde{x}-\langle x\rangle_{q_\theta(x|\tilde{x})})k(\tilde{x}|x)q_\theta(x)}{\tilde{q}_\theta(\tilde{x})}\,\mathrm{d}x$$

$$\implies \sigma^4\nabla^2_{\tilde{x}}\log\tilde{q}_\theta(\tilde{x}) + \sigma^2 I = \int\left(-(\tilde{x}-x)^2 + (\tilde{x}-x)(\tilde{x}-\langle x\rangle_{q_\theta(x|\tilde{x})})\right)q_\theta(x|\tilde{x})\,\mathrm{d}x$$

$$= \langle x^2\rangle_{q_\theta(x|\tilde{x})} - \langle x\rangle^2_{q_\theta(x|\tilde{x})} \equiv \Sigma_q(\tilde{x})$$

Therefore, we obtain the analytical full covariance identity.

$$\Sigma_q(\tilde{x}) = \sigma^2\nabla_{\tilde{x}}\mu_q(\tilde{x}). \quad (26)$$

### A.4 Proof of Theorem 2.3

**Lemma A.2** (KL to Gaussian [2]). *Let $p(x)$ be a distribution with mean $\mu_p$ and covariance $\Sigma_p$ and $q(x) = \mathcal{N}(\mu_q, \Sigma_q)$, denote the differential entropy as $\mathrm{H}(p) \equiv -\int p(x)\log p(x)\,\mathrm{d}x$, we have*

$$\mathrm{KL}(p||q) = \mathrm{KL}(\mathcal{N}(\mu_p, \Sigma_p)||q) + \mathrm{H}(\mathcal{N}(\mu_p, \Sigma_p)) - \mathrm{H}(p) \quad (27)$$

The proof can be found in [2] Lemma 2.

We can then prove Theorem 2.3. Since $p(\tilde{x}|x)p_d(x) = p(x|\tilde{x})\tilde{p}_d(\tilde{x})$, where $\tilde{p}_d(\tilde{x}) = \int p_d(x)p(\tilde{x}|x)\,\mathrm{d}x$, we have

$$\mathrm{KL}(p(\tilde{x}|x)p_d(x)||q(x|\tilde{x})\tilde{p}_d(\tilde{x})) = \langle\mathrm{KL}(p(x|\tilde{x})||q(x|\tilde{x}))\rangle_{\tilde{p}(\tilde{x})} \quad (28)$$

Assume Gaussian distribution $q(x|\tilde{x}) = \mathcal{N}(\mu_q(\tilde{x}), \Sigma_q(\tilde{x}))$ and denote the mean and covariance of the true posterior are $\mu_p(\tilde{x})$ and $\Sigma_p(\tilde{x})$, then the optimal $q^*$ is

$$q^* = \arg\min_q \mathrm{KL}(p(\tilde{x}|x)p_d(x)\|q(x|\tilde{x})\tilde{p}_d(\tilde{x})) \tag{29}$$

$$= \arg\min_q \left\langle \mathrm{KL}(p(x|\tilde{x})\|q(x|\tilde{x})) \right\rangle_{\tilde{p}(\tilde{x})} \tag{30}$$

$$= \arg\min_q \left\langle \mathrm{KL}(\mathcal{N}(\mu_p, \Sigma_p)\|q(x|\tilde{x})) + \mathrm{H}(\mathcal{N}(\mu_p, \Sigma_p)) - \mathrm{H}(p(x|\tilde{x})) \right\rangle_{\tilde{p}(\tilde{x})} \tag{31}$$

$$= \arg\min_q \left\langle \mathrm{KL}(\mathcal{N}(\mu_p, \Sigma_p)\|q(x|\tilde{x})) \right\rangle_{\tilde{p}(\tilde{x})} + const.. \tag{32}$$

Therefore, the optimal $q(x|\tilde{x}) = \mathcal{N}(\mu_q(\tilde{x}), \Sigma_q(\tilde{x}))$ under the joint KL has the mean and covariance $\mu_q^*(\tilde{x}) = \mu_p(\tilde{x}), \Sigma_q^*(\tilde{x})) = \Sigma_p(\tilde{x})$.

## B Connection to Analytical DDPM

Paper [2] considers the constrained variational family $q_\theta(x|\tilde{x}) = \mathcal{N}(\mu_\theta(\tilde{x}), \sigma_q^2 I)$ and derive the optimal $\sigma_q^*$ as

$$\sigma_q^{*2} = \arg\min_{\sigma_q} \mathrm{KL}(p(\tilde{x}|x)p_d(x)\|q_\theta(x|y))\tilde{p}_d(\tilde{x})) = \frac{1}{d} \left\langle \mathrm{Tr}\left(\mathrm{Cov}_{q(x|\tilde{x})}[x]\right)\right\rangle_{\tilde{p}_d(\tilde{x})}, \tag{33}$$

which can also be rewritten using the score function

$$\sigma_q^{*2} = \sigma^2 - \frac{\sigma^4}{d} \left\langle \|s_{q_\theta}(\tilde{x})\|_2^2 \right\rangle_{\tilde{p}_d(\tilde{x})}. \tag{34}$$

To make a deep connection, we can also plug our analytical full covariance (Equation 10) into Equation 16

$$\sigma_q^{*2} = \sigma^2 + \frac{\sigma^4}{d}\mathrm{Tr}\left\langle \nabla_x^2 \log q_\theta(\tilde{x})\right\rangle_{\tilde{p}_d(\tilde{x})}$$

$$= \sigma^2 - \frac{\sigma^4}{d}\mathrm{Tr}\left\langle s_{q_\theta}(\tilde{x})s_{q_\theta}(\tilde{x})^T\right\rangle_{\tilde{p}_d(\tilde{x})} = \sigma^2 - \frac{\sigma^4}{d}\left\langle \|s_{q_\theta}(\tilde{x})\|_2^2\right\rangle_{\tilde{p}_d(\tilde{x})}, \tag{35}$$

which recovers Equation 17, where the first equality is due to the well-known Fisher information identity [10].

## C Experiments

All the experiments conducted in this paper are run on one single NVDIA GTX 3090.

### C.1 Effect of the Single Noise Choice on MNIST

Figure 11 shows the samples generated by our method with the EBM trained with difference $\sigma \in \{0.3, 0.5, 0.8\}$ in the noise distribution $p(\tilde{x}|x)$, we can find the image quality also heavily depends on the choice of the noise scale and $\sigma = 0.5$ achieves the best visual quality, we then use this hyper-parameter in the subsequent comparisons.

### C.2 Multi-level Noise Details

For full details on the architecture and noise schedule used in the multi-level noise experiments in Section 5, we refer to Appendix B of [36]. For our multi-level Gibbs sampling procedure, we used 3 Gibbs steps at each noise level and 3 Rademacher samples for each diagonal Hessian computation. Following [36], we used a total of 232 noise levels, distributed according to their proposed geometric schedule, and applied a final denoising step in which the mean of the clean distribution conditioned on the final output of the sampling procedure is returned (the final output of the sampling procedure is a sample from the noised distribution from the noise distribution at the smallest noise level). This denoising step was previously found to improve FID scores [17] significantly.

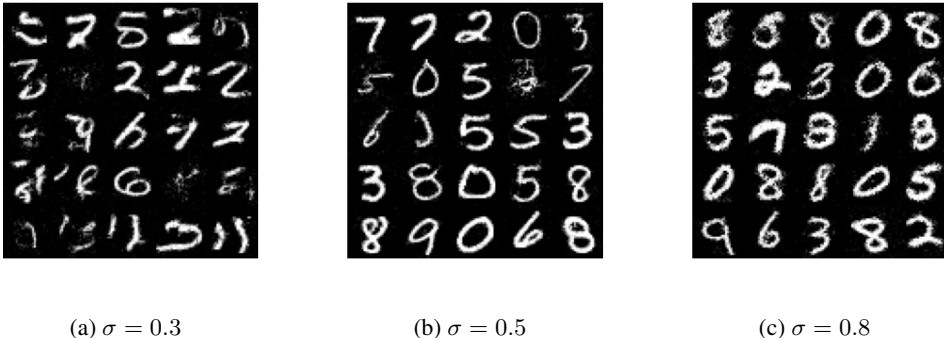

(a) $\sigma = 0.3$        (b) $\sigma = 0.5$        (c) $\sigma = 0.8$

Figure 11: Sample comparisons with different $\sigma$ value.

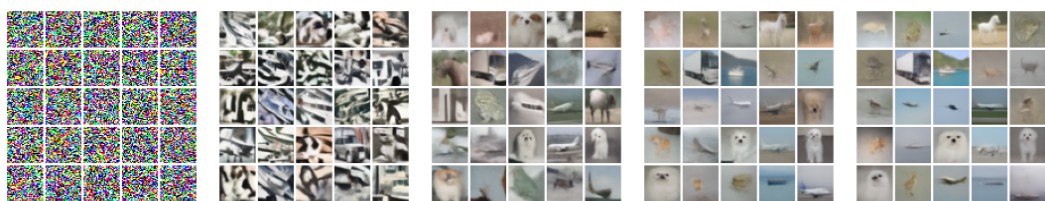

Figure 12: Mode Collapse visualization of 25 Markov chains, we plot the samples every 20 Gibbs steps, we can find less modes are covered if we run the Gibbs sampling for a longer time.

