# OpenReview forum: "Moment Matching Denoising Gibbs Sampling"
_NeurIPS.cc/2023/Conference — NeurIPS 2023 poster_

### Official Review · Reviewer_1SZs · 2023-06-26

**Soundness:** 3 good
**Presentation:** 3 good
**Contribution:** 2 fair
**Rating:** 6
**Confidence:** 3

**Summary:**

This paper proposes a new denoising scoring matching (DSM) technique to train energy-based models. They address the well-known inconsistency problem of DSM objectives by training a "noisy" model and then sampling "clean" samples from the "noisy" model by learning a denoising distribution.

The contributions of the paper are as follows:

1) Theoretically, they show that a "clean" model is uniquely identified by an EBM learning a noisy data distribution.
2) A sampling strategy inspired from Gibbs sampling to approximate the denoising distribution, which allows to sample from the underlying clean model without the need to retrain the noisy model.
3) Practically, they show how to scale their approach to high-dimensional data by overcoming the quadratic cost of evaluating the Hessian of the DSM for every $\tilde{x}$.

**Strengths:**

- Well-written paper, easy to follow and most claims are justified either by citing previous work or in appendix / math proofs
- Prior work is duly cited
- Experiments on synthetic and real-world (MNIST/CIFAR) datasets are appreciated to assess the quality of the technique proposed

**Weaknesses:**

I'm not an expert in this field and I have a hard time evaluating the overall contribution of this paper to the field of "Generative Models". In particular, it feels like this denoising techniques introduce a scalability problem (although partly discussed and solved by contribution 3) in order to solve the inconsistency problem of score-based matching training. A discussion on the quality of the approximation and comparing the full Hessian with its approximation could help answer my doubts.

Besides, I think that describing the full sampling algorithm could help the reader get a better picture of the whole training and sampling procedure. This would clarify what has been made in the experiments section.

I list below a few comments / typos I have found while reviewing the paper:

1) l.33: there is a missing $\nabla_x$ in the definition of the score matching function. Otherwise, I don't see how $\nabla_x s_{p_{\theta}}(x) = -\nabla^2_x f_{\theta}(x)$.
2) l.26: "it is equivalent to MLE". A reference is welcome.
3) l.34: "under certain regularity conditions, the Fisher divergence is equivalent to SM". Reference?

**Questions:**

As stated above in the weaknesses section, my main doubt revolves around how does the Hessian approximation affect the training stability and overall output samples quality.

**Limitations:**

The authors have partly discussed the limitations in section 3, but an experiment comparing the quality of the approximation of the generated samples with the full Hessian and the approximated Hessian is welcome to strengthen the demonstration.

No potential negative societal impact at this stage.

---

> ### Author Rebuttal · Authors · 2023-08-09
>
> We're grateful for the constructive feedback.  We're glad the clarity, references, and experiments in our paper resonated with you. Your feedback greatly enhances the paper's quality.
>
> ### Typos and Reference
> We want to thank the reviewer for pointing out the typo $s(x)\equiv \nabla_x\log p(x)$. The reference for the equivalent of MLE and KL training can be found in [1] and the connection between the Fisher divergence and the Score matching can be found in [2], which we will include in the revised version.
>
> ### Scalability
> To demonstrate our method's scalability, we further conduct an experiment using the CelebA dataset, characterized by its dimensions of $64{\times}64{\times}3$. For visual evidence of the generated samples, kindly refer to the PDF in the global rebuttal. The results highlight our method's capacity to produce high-resolution, quality human faces.
>
> ### Stability and Approximation Quality
> Furthermore, we would like to provide a clear explanation regarding our methodology: the estimation of the Hessian is required only at the time of generation, not during training. The training process adheres to the traditional framework of denoising score-matching. As visualized in Figure 5, our depiction of covariance estimation with different Rademacher samples will be supplemented in the revised version with a study on how varying samples might influence the quality of the results. We deeply appreciate this insightful suggestion, and it will certainly enhance the comprehensiveness of our research.
>
> ### Reference
> [1]: Bishop C M, Nasrabadi N M. Pattern recognition and machine learning. New York: springer, 2006.
>
> [2]: Hyvärinen A, Dayan P. Estimation of non-normalized statistical models by score matching. Journal of Machine Learning Research, 2005.

---

### Official Review · Reviewer_afYY · 2023-07-05

**Soundness:** 3 good
**Presentation:** 3 good
**Contribution:** 3 good
**Rating:** 5
**Confidence:** 1

**Summary:**

This paper proposed a sampling scheme called (pseudo)-Gibbs sampling with moment machine. It can be applicable to the sampling problem from  from the underlying clean model when a noisy model hat has been well-trained via DSM. The proposed sampling technique is used to sample images for  image generation and showed good efficiency on toy dataset.

**Strengths:**

1. The paper is well written
2. The problem is a basic problem with many applications
3.  The idea is new and seems working well.

**Weaknesses:**

The improvement in image generation seems minor.

**Questions:**

NIL

**Limitations:**

NIL

---

> ### Author Rebuttal · Authors · 2023-08-09
>
> We appreciate the reviewer for acknowledging the strengths of our paper.
>
> ### Contribution and Effectiveness
> We wish to emphasize that this paper's central contribution is to elucidate the theoretical optimal full-covariance Gaussian approximation for the denoising distribution. As delineated in Theorems 2.2 and 2.3, our proposed approximation stands as theoretically optimal. In practical applications, our approach offers enhanced performance in single-noise level Gibbs sampling, all while reducing both parameter needs and training expenses—particularly, our method obviates the necessity of learning the covariance.
>
> In the context of multi-level noise, we aim to position our Gibbs sampler as a viable alternative to the prevalent Langevin dynamics method within the score-based diffusion model. Our experiment on the CelebA dataset (64x64x3) bolsters this claim, illustrating our method's proficiency in generating high-quality images. For a glimpse of these visuals, kindly refer to the accompanying PDF in the comprehensive review. Such findings effectively echo our foundational aspirations. We are inclined to believe that a deeper dive into the interplay between denoising Gibbs sampling and Langevin dynamics MCMC could shed additional light, further refining the diffusion model's sampling process. This presents a promising avenue for future exploration.

---

### Official Review · Reviewer_DnqP · 2023-07-07

**Soundness:** 3 good
**Presentation:** 3 good
**Contribution:** 2 fair
**Rating:** 5
**Confidence:** 4

**Summary:**

This paper addressed the inconsistency problem in training energy-based models (EBMs) using denoising score matching (DSM). An efficient sampling framework, called (pseudo)-Gibbs sampling with moment matching, was proposed to enable effective sampling from the underlying clean model when given a well-trained 'noisy' model obtained through DSM. The presence of an underlying clean model within the noisy EBM was identified, and a pseudo-Gibbs sampling scheme incorporating analytical moment matching approximation was introduced for sampling from the clean model without requiring additional training. The effectiveness of the method was demonstrated in generating high-quality images for both single and multi-level noise settings. Additionally, the scalability of the method to high-dimensional datasets was illustrated. Comparisons with related methods highlighted the advantages of the proposed approach.

**Strengths:**

The theoretical derivation and presentation are clear and well-defined.

**Weaknesses:**

The limited number of experiments conducted to validate the proposed method. While the paper provides some demonstration of the effectiveness of the approach, further experiments across a wider range of scenarios and datasets would enhance the robustness and generalizability of the findings.

The authors should provide more details for those experiments in Supplementary Material.

The authors failed to demonstrate the wide applications of the proposed method.

**Questions:**

In experiments, the authors used a noise-conditioned score network trained in [32], does the proposed model also excel in other settings?

---

> ### Author Rebuttal · Authors · 2023-08-09
>
> We're grateful to the reviewer for their valuable feedback, which has undeniably enhanced the quality of our work. The commendation on our theoretical contribution and presentation is heartening and much appreciated. Below, we respond to the queries presented.
>
> ### Wider scenarios and applications
> We want to thank the author for the feedback, emphasizing the need for broader experimentation to solidify the robustness and generalizability of our findings. In response to this, we have extended our methodology to include the CelebA dataset, notable for its heightened dimensionality (64x64x3), see the attached PDF in the global rebuttal for the sample results. The generated samples further highlight our method's capability to produce high-quality human face images.
>
> In addition to the image generation setup, our method is generally applicable for any continuous Energy-based models or denoising score matching training, see [1] for a reference of different energy-based model applications. We will also include a discussion of wider applications in our revised paper.
>
> ###  Experimental details
>  The suggestion to provide more detailed information about the experiments in the Supplementary Material is well-received. We will include expanded details and the code in the revised paper.
>
>
> ###  Applicability in other diffusion models
>  Our proposed method boasts a versatile application spectrum, suitable for any diffusion model employing a forward Gaussian convolution process. This includes but is not limited to, the hierarchy latent variable model and the score-based model perspective. The profound interrelation between these perspectives is elaborated upon in [2]. We appreciate the reviewer's suggestion and will incorporate a detailed discussion on this topic in the updated version of our paper.
>
> ### Reference
> [1] Luo C. Understanding diffusion models: A unified perspective. arXiv preprint arXiv:2208.11970, 2022.

---

> > ### Comment · Reviewer_DnqP · 2023-08-18
> >
> > After considering the rebuttal provided in response to my feedback, I find that the authors have addressed the majority of the raised concerns and have committed to enhancing their experiments. Consequently, I maintain my rating of Borderline accept.

---

### Official Review · Reviewer_oYaT · 2023-07-07

**Soundness:** 3 good
**Presentation:** 2 fair
**Contribution:** 3 good
**Rating:** 5
**Confidence:** 3

**Summary:**

This work proposes a sampling method from implicitly defined clean model given a noisy model without retraining and demonstrates its applicability to high-dimensional data / images.

**Strengths:**

- this work presents interesting theoretical results such as the uniqueness of underlying clean model given noisy data distribution, analytical covariance identity, and optimal Gaussian approx.
- this work presents interesting corrections to prior works such as analytical DDPM, KL training Gibbs sampling.

**Weaknesses:**

- It is unclear if the proposed method (e.g., Eq. (12)) is effective in general. Some of the toy examples look promising, but some image examples may not look good (see Table 2 for large performance gaps with DDPM and NCSN++ / possibly more with recent methods). While this work emphasized that it will work well with a single level of fixed noise, eventually the proposed method with multi-level noise was used for more complex image cases without the issue of mode collapse. This inconsistency is my major concern.
- Weak experimental results: some of the experiments may not effectively demonstrate the performance of the proposed scheme well. For example, it is unclear what this work wanted to show in Figures 5, 6 and 8. Comparison experiments may be needed.


**Questions:**

Please address the comments in the Weaknesses section.

- What if log scale is used in Fig 3(a)? Then, it may show similar curves like Fig 3(b) unlike the claims in this work.
- It is unclear why the comparable results of NCSNv2 with Langevin / the proposed method is ok. Are they using the same multi-level noise? Then, what is the advantage of the proposed method over the conventional Langevin dynamics?

**Limitations:**

Yes

---

> ### Author Rebuttal · Authors · 2023-08-09
>
> We express our gratitude to the reviewer for recognizing the depth and value of our research. It's gratifying to see the acknowledgement of our novel theoretical findings. Here are our responses to the questions posed.
>
> ### Effectiveness of the Proposed Method
> We wish to emphasize that the central thrust of this paper is the elucidation of the optimal full-covariance Gaussian approximation within the denoising distribution. As outlined in Theorems 2.2 and 2.3, we've shown the theoretical superiority of our proposed approximation. Practically, our approach offers enhanced outcomes in single-noise level Gibbs sampling, achieving this with reduced parameter demands and training overhead, eliminating the need to learn the covariance.
>
> For multi-level noise, our goal is to show that the proposed Gibbs sampler can be a competitive drop-in replacement to the widely used Langevin dynamics method in the score-based diffusion model. We also conduct an experiment on CelebA ($64{\times}64{\times}3$) dataset, which further shows that our method can generate high-quality images, see the PDF in the global rebuttal for the sample visualizations. Overall, these findings serve as a successful validation of our proposed proof of concept.
>
> In reference to Table 2, our results are derived from a pre-trained NCSN v2, which exhibits performance on par with Langevin dynamics sampling. This comparison is fair (we use the same pre-trained network and noise schedule) and substantiates our experimental objective. Discrepancies in performance between other diffusion models can be attributed to variations in architectures and training procedures. The adaptation of these models in conjunction with our method represents a promising avenue for future research.
>
>
> ### Inconsistency between applications
> The paper's primary contribution lies in deriving the theoretical optimal Gaussian moment matching for Gaussian denoising distribution. We demonstrate that the proposed approximation is applicable in both single-noise and multi-level noise scenarios, treating them as separate applications. This ensures the proposed moment-matching technique is consistent across varied contexts. We appreciate the reviewer's feedback on this matter and will clarify further on this aspect in the revised manuscript.
>
>
>
> ### Interpretation of the figures:
> We are grateful to the reviewer for highlighting the need for a clearer objective for the figures presented. We will add the following clarification in our revised paper:
>
> - Figure 5 offers a visualization of the covariance estimation quality with different Rademacher samples, illustrating a discernible trade-off between estimation quality and computational cost.
> - Figure 6 portrays samples from three independent Gibbs Chains trained with single-level noise, adhering to standard plot conventions in MCMC research, and proving that high-quality images can be generated with only fixed single-level noise.
> - Figure 8 presents samples from the multi-level noise scenario, providing further insight into this specific case.
> In the revised version of the paper, we will include more detailed clarifications for these figures, and we extend our gratitude to the reviewer for drawing attention to this need.
>
> ### Figure comparison 4a vs 4b:
> Figure 4a depicts the diagonal covariance estimated using our method, while Figure 4b represents the diagonal covariance obtained through KL minimization. As shown in Theorem 2.3, the optimal diagonal covariance learned by KL (Figure 4b) should resemble that in Figure 4a, this is why they should look similar. However, the crucial distinction is that our method (Figure 4a). directly estimates the optimal covariance without the necessity for training. In contrast, the KL learning approach (Figure 4b) necessitates additional parameters and training, potentially leading to sub-optimal covariance estimation. This is further illustrated in Figures 4d and 4f, demonstrating that our method yields superior sample generation compared to the covariance learning approach. We'll provide a more detailed clarification in our revised paper.

---

> > ### Comment · Reviewer_oYaT · 2023-08-20
> >
> > I would like to thank the authors for detailed responses on my concerns.

---

### Official Review · Reviewer_ddyX · 2023-07-27

**Soundness:** 3 good
**Presentation:** 2 fair
**Contribution:** 2 fair
**Rating:** 6
**Confidence:** 3

**Summary:**

This paper proposes a sampling framework, i.e., (pseudo)-Gibbs sampling with moment matching, which aims at sampling from the implicitly-defined clean model given the noisy energy-based model well-trained by the denoising score matching method. It first provides a theoretical justification of the existence of the underlying clean model for a well-trained energy-based model. It then introduces the gibbs sampling framework based on the proposed analytical Gaussian moment matching approximation method for the denoising distribution. Experiments on several toy datasets and MNIST and CIFAR10 datasets are conducted to demonstrate the effectiveness of the proposed approximation and sampling method.

**Strengths:**

- The paper provides theoretical justification of the existence of the underlying clean model of a well-trained noisy EBM, and discusses the existence of the clean model for an imperfect EBM.
- It derives an analytical Gaussian moment matching approximation method for the denoising distribution and validates the design on several toy datasets, and discusses some scalable implementations of the proposed method for image data.
- The presentation in general is clear.

**Weaknesses:**

- The full covariance moment matching requires calculating the Hessian matrix, which could be computationally expensive even with diagonal approximation methods.
- The experiments are conducted on toy datasets and MNIST and CIFAR-10 datasets. More results on other image dataset could provide better intuitive understanding of the proposed method.
- The performance of the proposed sampling method is close to the existing methods. The significance of the proposed sampling method is not very clear.

**Questions:**

Please see the weaknesses section.

**Limitations:**

The authors have mentioned the limitations in Sec 3. Scalable Implementations for Image Data and Sec 4. Image Generation with a Single Noise Level.

---

> ### Author Rebuttal · Authors · 2023-08-09
>
> We thank the reviewer for their time and valuable feedback that improves the quality of our work. We are encouraged by the positive comments regarding our theoretical contribution and presentation.
>
> ### Hessian computation
> We want to highlight that our method can directly obtain the optimal covariance by **only learning the mean function** during training. In contrast, the classic KL training Gibbs Sampling approach necessitates the learning of **both the mean and covariance functions** in the training time. This not only demands extra parameters and increases computational costs but also poses a risk of yielding a sub-optimal solution. Therefore, our method is both **theoretically optimal** and **more computationally efficient in the training stage** (our method doesn’t require any Hessian computations in the training).
>
> However, in the generation stage, we do agree with the reviewer that calculating the Hessian or even diagonal Hessian is computationally expensive for high-dimensional data, but it completes the following spectrum of **flexibility and efficiency trade-offs**:
>
> - Data-independent isotropic Gaussian approximation proposed [1]: This is theoretically the least flexible approach but is the most computationally efficient approach.
> - Proposed data-dependent Diagonal Gaussian approximation: More flexible than the data-independent isotropic method, but less computationally efficient.
> - Proposed data-dependent full covariance Gaussian approximation: This method represents the theoretical optimal point on the flexibility spectrum, but is the least computationally efficient of the three options.
>
> We posit that these trade-off methods will expand the repertoire of ML toolboxes, catering to various applications, each with its distinct demands for accuracy and efficiency.
>
>
> ### Additional Experiments
> We appreciate the reviewer's suggestion to test our method on an additional dataset.  In response to this, we have extended our methodology to include the CelebA dataset, notable for its heightened dimensionality (64x64x3), see the attached PDF in the global rebuttal for the sample results. The generated samples further highlight our method's capability to produce high-quality human face images, other experimental details and numerical evaluations will be added to the revised version.
>
> ### Contribution and Effectiveness
>  We would like to clarify that the primary contribution of this paper is to unravel the theoretical optimal full-covariance Gaussian approximation of the denoising distribution. This is an open problem in the realm of the denoising score-matching method. The recent paper [1] preliminary derived an analytical form in a constrained data-independent isotropic  Gaussian approximation setting. Our investigation has led to a surprising discovery that this optimal approximation can be elegantly expressed in an analytical form for the full-covariance data-dependent moment matching, providing a solution that is both more flexible and accurate.
>
> We've demonstrated that, in the single-noise level setting, our proposed method **outperforms** the classic denoising Gibbs sampling, and it does so with fewer parameters and lower training costs. For multi-level noise, our primary goal is to provide a proof of concept that the proposed Gibbs sampler can serve as a **competitive drop-in replacement** for the widely used Langevin dynamics MCMC method in the score-based diffusion model.  Our results have successfully validated this goal. We posit that delving into the relationship between denoising Gibbs sampling and Langevin dynamics MCMC will deepen our comprehension and refine sampling within the diffusion model. This exploration stands as an enticing avenue for future research.
>
>
> ### Reference
> [1] Bao F, Li C, Zhu J, et al. Analytic-dpm: an analytic estimate of the optimal reverse variance in diffusion probabilistic models. ICLR, 2022.

---

> > ### Comment · Reviewer_ddyX · 2023-08-19
> > **Thanks for Your Response**
> >
> > I thank the authors for their response and the effort they put in resolving my concerns. I strongly recommend the authors to incorporate the new results in the rebuttal especially those on new datasets into the next version of the paper, as I believe these will further strengthen the paper.
> >
> > In light of the new results and the responses from the authors, I will increase my score to 6.

---

### Author Rebuttal · Authors · 2023-08-09

We want to thank all the reviewers again for the time spent and their valuable feedback that improves the quality of our work. Their recognition of our theoretical contribution, which forms the cornerstone of our paper, is deeply encouraging. Here's a succinct overview of our primary contributions.

### Contributions
1. We introduced an analytical data-dependent full covariance moment-matching objective for the posterior within the Denoising Score Matching framework and validated its **theoretical optimality**.
2. We pinpointed the **existence of the clean model** in a noisy EBM trained via denoising score matching and developed a moment-matching Gibbs sampler for the energy-based models. This approach drastically cuts down on parameters, training expenses, and boosts the performance of the conventional denoising Gibbs sampling [1,2] with amortized covariance learning [3].
3. We further scale moment-matching estimation for high-dimensional data, which provides a **trade-off between flexibility and efficiency**.  We also demonstrate the scalable method can:
     - generate high-quality natural images with even **one fixed-level noise**.
     - achieve **competitive performance** compared to the widely used Langevin Dynamics sampling in multiple-noise settings.

### Effectiveness
A shared concern among the reviewers centres on the effectiveness of the proposed method. We want to highlight that, as mentioned in contribution point 2, we've demonstrated that our method not only yields enhanced performance in single-noise level settings compared to traditional denoising Gibbs sampling but also achieves this with reduced training costs, notably eliminating the need to learn the covariance.

For multi-level noise, our goal is to provide a proof of concept that the proposed Gibbs sampler can serve as a competitive drop-in replacement for the widely used Langevin dynamics method in the score-based diffusion model. Our preliminary results have successfully validated this goal for CIFAR10. Since submission, we have run further experiments on the CelebA dataset, demonstrating scaling to image sizes $64{\times}64{\times}3$, see the attached PDF for representative samples. We can see our method is also able to generate high-quality images in a higher dimension space. Experimental details and numerical evaluations will be added in the camera-ready version.

### Reference
[1] Bengio Y, Yao L, Alain G, et al. Generalized denoising auto-encoders as generative models. Advances in neural information processing systems, 2013, 26.

[2] Alain G, Bengio Y. What regularized auto-encoders learn from the data-generating distribution. The Journal of Machine Learning Research, 2014

[3] Kingma D P, Welling M. Auto-encoding variational bayes. arXiv preprint arXiv:1312.6114, 2013.

---

### Decision · Program_Chairs · 2023-09-21

**Decision:**

Accept (poster)

**Comment:**

This paper proposes a new sampling framework, i.e., (pseudo)-Gibbs sampling with moment matching, for the sake of drawing samples from the implicitly-defined clean model given the noisy energy-based model, which is well-trained via denoising score matching method. The paper provides both theoretical and empirical justification of the proposed method.  In response to the concerns initially raised by the reviewers in their initial assessments, the rebuttal systematically addressed these issues. Following extensive discussions, a consensus was achieved among all five reviewers, culminating in the decision to accept the paper. This agreement is attributed to the paper's profound theoretical justification, and sufficient empirical validation that it offers for this particular study. The AC agrees with the reviewers and recommends accepting the paper. To further improve the paper quality, AC suggests the authors to revise their paper by taking into account all the suggestions provided by the reviewers, including integrating extra experimental results conducted during the rebuttal phase.